# Conventional vs. Organic: Evaluation of Nutritional, Functional and Sensory Quality of *Citrus limon*

**DOI:** 10.3390/foods12234304

**Published:** 2023-11-28

**Authors:** Paola Sánchez-Bravo, Juan Martínez-Tomé, Francisca Hernández, Esther Sendra, Luis Noguera-Artiaga

**Affiliations:** 1Research Group “Food Quality and Safety”, Instituto de Investigación e Innovación Agroalimentaria y Agroambiental (CIAGRO), Miguel Hernández University of Elche (UMH), Carretera de Beniel Km 3.2, 03312 Orihuela, Spain; paola.sb94@gmail.com (P.S.-B.); esther.sendra@umh.es (E.S.); 2Department of Plant Sciences and Microbiology, Instituto de Investigación e Innovación Agroalimentaria y Agroambiental (CIAGRO), Miguel Hernández University of Elche (UMH), Carretera de Beniel Km 3.2, 03312 Orihuela, Spain; juan.martinez@umh.es (J.M.-T.); francisca.hernandez@umh.es (F.H.)

**Keywords:** amino acids, descriptive analysis, ‘Fino 49’, lemon fruits, polyphenols

## Abstract

Organic farming is growing rapidly worldwide since it is perceived as more respectful of the environment than conventional farming. In this sense, organic agriculture is highly appreciated by consumers since consumers around the world believe that organic food has a higher content of beneficial compounds for health and consider it of higher quality. For that reason, the objective of this research was to evaluate the nutritional, sensorial, and functional quality of the ‘Fino 49’ lemon grafted on *Citrus macrophylla* in conventional and organic cultivation. Fatty acids, amino acids, total phenol, and polyphenols were quantified, antioxidant activity was measured, and sensory descriptive analysis was performed. Conventional farming led to an increase in amino acid content (641 mg L^−1^) and an increase in polyunsaturated fatty acids (254 mg 100 g^−1^) and monounsaturated fatty acids (37.61 mg 100 g^−1^). On the other hand, organically produced lemon fruits had better sensory profile (highlighting overall aroma (6.5), lemon odor (6.8), sourness (5.8), floral (0.6), and fresh lemon flavor (9.8)), and lower thrombogenicity index (0.15). The type of cultivation (organic and conventional) had no influence on the antioxidant activity (~1.60, ~3.08, and ~4.16 mmol Trolox L^−1^ for ABTS+, DPPH•, and FRAP, respectively) and polyphenols content (85.51 and 86.69 conventional and organic, respectively). However, to establish the advantages and disadvantages of different types of cultivation on lemon quality more studies are needed.

## 1. Introduction

Organic farming emerged in the 1920s as a result of protests towards the excessive industrialization of agriculture [1]. Organic agriculture combines traditional agriculture with modern agricultural technologies, promoting techniques that take advantage of soil fertility and biodiversity, such as crop rotation, crop diversification, or pest management naturally [2,3,4,5].

Today, the demand for organic foods is growing rapidly worldwide [6]. However, organic farming is considered an inefficient approach to agriculture as it sometimes produces lower yields compared to conventional agriculture under the same conditions [6,7]. On the other hand, organic farming is more profitable and environmentally friendly, producing equally or more nutritious food [6]. In general, it can be said that organic orchards are more environmentally friendly than conventional ones. Organic farming reduces the application of fertilizers (minerals and pesticides) in its production, but it does not completely eliminate them since it still uses non-synthetic pesticides (sulfur, copper sulfate, potassium permanganate, etc.) [3]. Nowadays, the extension of cultivated areas due to the increase in organic farming has increased notably, reaching almost 14 million hectares of organic farming in the European Union in 2019 [8] due to a conversion of conventional farming to organic farming. Furthermore, a larger budget has also been allocated to organic production [9].

In general, organic agriculture presents soils with a higher content of organic matter [10], although yields may be lower depending on the characteristics of the crop and soil [11]. Recent studies have shown that, under identical soil conditions, after the initial years of cultivation, the yields of organic systems equaled those of conventional ones, requiring lower nitrogen inputs, having greater stability (mineralization of nutrients, pH, and soil biota), and better soil structure (organic matter and nitrate reduction in groundwater) [12]. However, how the growing method (conventional or organic) affects the quality of the fruit/vegetable is still difficult to establish. Maggio et al. [13] established that endive, zucchini, and cauliflower cropping systems influence differently depending on soil type and crop variables. Therefore, understanding how these variables affect the final quality is key to improving a sustainable and efficient farming system.

Lemon is an important crop worldwide [14], with the main producers being India, Mexico, and China, respectively [15]. In Spain, one of the 10 main lemon producers, especially in Mediterranean areas, 884,890 tons of lemon were produced in 2019 [15]. The varieties most produced in this country are “Verna” and “Fino” with more than 98% of the cultivated area being the cultivar ‘Fino 49’, the one that has the greatest projection among Spanish farmers [16].

In recent years, many citrus orchards have been transformed into organic farming [4]. In fact, lemon trees are well adapted to organic cultivation, being able to obtain a high harvest [17]. In Spain, the organic lemon cultivation area grew by almost 16% in 2018, from 12,087 to 14,017 hectares nationwide [18]. In addition, in Spain, more than 85% of the consumption of organic foods is in vegetables and fruits [19].

Currently, the quality of the fruit is valued, not only at the visual level (size, color) but also at the nutritional and functional level (content of carotenoids, minerals, phenols, vitamins, and volatile compounds, etc.) [14,20,21,22,23]. In this sense, organic agriculture is highly appreciated by consumers since consumers around the world think that organic food has a higher content of beneficial compounds for health and consider it of higher quality [24], although it has not currently been supported by scientific data [25]. Recently, it has been shown that environmental issues are the main factor in the choice of purchase of organic products by consumers [26], although it is true that some authors also established that the decision to buy organic foods is due to the beliefs of consumers about the health benefits that organic foods represent over the environmental benefits [25,27].

For all these reasons, the objective of this research was to evaluate the nutritional, sensorial, and functional quality of the Fino 49 lemon grafted on *Citrus macrophylla* in conventional and organic cultivation to determine if the farming conditions influence its composition.

## 2. Materials and Methods

### 2.1. Plant Material

Fino 49 variety grafted on *Citrus macrophylla* was used for the study. The cultivation of organic lemon was carried out in the “Lo Vera” orchard (10.28 ha; 38°03′24.6″ N, 0°44′25.0″ W) and the “Lo Lorente” orchard was used for conventional cultivation (9.11 ha; 38°02′21.3″ N, 0°44′51.3″ W). The orchard conditions were fully described by Sánchez-Bravo et al. [28]. Briefly, organic and conventional orchards were in Alicante, Spain with a plantation framework of 6.5 × 4 m. Organic orchard had clay-textured soil, with an organic matter content of ~1.40% and conventional orchard had loam-clay texture soil, with <0.65% of organic matter content.

Samples were harvested in October 2021. One hundred fruits of each treatment were collected from the middle part of the tree. Twenty-five fruits were used for the descriptive sensory analysis and the other 75 were used for the rest of the analysis.

### 2.2. Fatty Acids

Fatty acid content was determined by the fatty acid methyl ester method (FAME). Lyophilized lemon husk fat was directly methylated according to Trigueros et al. [29]. For identification and quantification, a Nexis GC-2030 (Shimadzu Scientific Instruments, Inc., Columbia, MD, USA) gas chromatograph (GC) coupled with a flame ionization detector (FID) with AOC-20i injector was used to analyze fatty acids methyl esters, using helium as a carrier gas and air and hydrogen as detector gases. The GC column was a Supelco SP^®^-2380 Capillary GC, with a length of 60 m, an internal diameter of 0.25 mm, and a film thickness of 0.20 μm. The temperature of the injector was 250 °C with a 1:20 split ratio. The linear velocity of the carrier gas was 28.4 cm s^−1^ and the total column flow was 1.58 mL min^−1^ (He). The oven program had a temperature ramp of 3 °C min^−1^, from 70 °C to 250 °C. The detector temperature was 260 °C using H_2_ and air at 32 mL min^−1^ and 200 mL min^−1^, respectively; and N_2_ was make up at 24 mL min^−1^. For the identification of fatty acids, a mixture of FAME (18912-1AMP, Sigma-Aldrich, Saint Louis, MO, USA) was used, and for the quantification, C19:0 (0.1 mg mL^−1^) was used as internal standard. Results were expressed as mg 100 g^−1^.

### 2.3. Antioxidant Activity and Total Polyphenol Content

The antioxidant activity (AA) and total polyphenol content (TPC) were determinate in lemon samples. Samples were extracted according to Noguera-Artiaga et al. [30], with some modification. Briefly, 1 mL of lemon juice was mixed with 10 mL of MeOH/water (80:20, *v*/*v*) in 1% HCl, sonicated for 15 min (20 °C), left for 24 h at 4 °C, and sonicated again for 15 min. Then, the extract was centrifugated for 10 min at 10,000 rpm. For the determination of TPC, the Folin-Ciocalteu reagent was used, following the technique described by Singleton et al. [31]. Antioxidant activity was measured by three methods, ABTS+ [32], FRAP [33], and DPPH• radical [34]. A UV-visible spectrophotometer (Thermo Fisher Scientific Helios Gamma model, UVG 1002E, Waltham, MA, USA) was used for the determination of TPC (765 nm) and antioxidant activity. Results were expressed as mg of gallic acid equivalents (GAE) per 100 mL^−1^ of lemon juice for TPC and as mmol Trolox L^−1^ of lemon juice for antioxidant activity. A concentration from 0.5 to 1.0 g GAE L^−1^ was used for the calibration curve (r^2^ = 0.999) for the quantification of total phenol content and 1.0 to 5.0 mmol Trolox L^−1^ for calibrations curves of AA (r^2^ ≥ 0.998). All the analyses were conducted in triplicate.

### 2.4. Amino Acids

The determination of amino acids was conducted using the method described by Kıvrak et al. [35] with some modifications. In this case, since the samples were liquid, the samples were diluted with H_2_O Milli-Q, then vortexed for 3 min, centrifuged, and the supernatant filtered with a filter of 0.22 um pore size. The supernatant was injected into the UPLC-QToF-MS equipment for subsequent amino acid analysis. Results were expressed as mg L^−1^. Standards (Alanine, Arginin, Aspartic acid, Cystine, Glutamic acid, Glycine, Histidine, Isoleucine, Leucine, Lysine, Methionine, Phenylalanine, Proline, Serine, Threonine, Tryptophan, Tyrosine, and Valine) were purchased from Sigma–Aldrich.

### 2.5. Polyphenols

The determination of the polyphenols profile was performed using an HPLC-DAD-ESI-MSn Ion Trap (Agilent 1200 series). For the extraction, 1 mL of the juice was centrifuged at 12,000 rpm for 10 min and subsequently filtered by 0.45 µm. For the analysis of the sample, an HPLC with a 1200 series DAD detector was used, coupled to a mass spectrometer (ion trap and electrospray ionization interface (ESI)). The capillary temperature was 350 °C and the voltage was 3500 V for the ESI source. Helium was used as a collision gas (collision energy of 50%) and the fragmentation was performed in the ion trap. Data were acquired in negative ionization mode, and the mass range for precursor ions and their fragmentation was 100–1000 *m*/*z*. The chromatographic separation was carried out on a Polar C18 column (Phenomenex) with a 5 µm internal particle diameter and a size of 250 × 4.6 mm. The mobile phases used were water: formic acid (95: 1, *v*:*v*) in channel A and acetonitrile in channel B at a flow rate of 1 mL min^−1^. The gradient was as follows: (i) 5% B up to 80% B (25 min); (ii) %B maintained for 10 min; (iii) %B increased to 99%; and, (iv) %B maintained 2 min. Twenty µL of the sample were injected. Pure standards and their maximum absorbance spectrum (290 nm, 320 nm, and 520 nm wavelengths) through chromatography (HPLC with a UV detector DAD and connected online to the mass spectrometer) were used for the quantification. The standards used were Chlorogenic Acid, Cyanidin-3-Glucoside, and Rutin (Sigma-Aldrich). Quantification is always possible even if the compound is unknown since we are giving absorbance responses and integrating it with a known area against the reference standard. The results of the concentration of polyphenols were expressed as mg L^−1^.

### 2.6. Sensory Analysis

Sensory descriptive analysis was carried out with a trained panel with more than 1000 h of experience (9 panelists; 4 men and 5 women; aged 25–68 years old). All the panelists were members of the Food Quality and Safety (CSA) research group (Universidad Miguel Hernández de Elche, Orihuela, Spain). Ten milliliters of lemon juice at 20 °C were provided to each judge, in odor-free covered cups and codified using 3-digit numbers. Between samples, the judges cleaned the palate using unsalted biscuits and fresh water. The room temperature was 23 ± 2 °C and it had natural and fluorescent light. Five sessions of 1 h had been for juice sensory analysis (all evaluated in each session). A numerical scale (from 0 to 10) was used for measuring the strength of sample attributes (0 = none and 10 = extremely strong, with 0.5 increments).

The lexicon employed during the five sessions was developed by Leksrisompong et al. [36] with some modifications (Table 1). The attributes were: overall aroma, lemon odor, fresh lemon, citric, floral, fruity, cleaner, sweet, sour, bitter, other, aftertaste, numbing, mouth coating, tooth etching, and astringent.

### 2.7. Statistical Analysis

A one-way analysis of variance (ANOVA) followed by a multiple-range test (Tukey’s) was carried out for the statistical analysis of data. The significant difference was defined as *p* < 0.05 (95% confidence interval). XLSTAT software (Addinsoft, version 2016.02.27444) was used for the statistical analysis.

## 3. Results and Discussion

### 3.1. Fatty Acids

Seventeen fatty acids (FAMEs) were identified and quantified in lemon peels (Table 2): nine were saturated (C16:0; C18:0; C15:0; C14:0; C17:0; C12:0; C10:0; C13:0; and C8:0), five were monounsaturated (C18:1c9; C18:1c11; C18:1t9; C16:1; and C17:1), and three were polyunsaturated (C18:3n3 alpha; C18:2n6c; and C18:3n6 gamma). Conventional samples had higher concentrations of the three main fatty acids, linoleic acid (126.29 mg 100 g^−1^), α-linolenic acid (125.63 mg 100 g^−1^), and palmitic acid (73.36 mg 100 g^−1^) than organic samples (60.07, 76.15, and 30.28 mg 100 g^−1^, respectively). Therefore, it was not surprising that conventional lemons had a higher content of saturated fatty acids (107.44 mg 100 g^−1^) than organic (50.09 mg 100 g^−1^). In the same way, conventional lemons had the highest concentration of monounsaturated fatty acids (37.61 mg 100 g^−1^) and polyunsaturated fatty acids (254.00 mg 100 g^−1^) than organic lemons (15.38 and 138.16 mg 100 g^−1^, respectively). It is important to mention that polyunsaturated fatty acids are very important for health since the human body does not synthesize them and they must be ingested through diet [37]. These results are different from those found by Duru and Enyoh [38], who reported a lower concentration of unsaturated acids and a higher concentration of saturated fatty acids in organic lemons. Other authors have reported no significant differences between conventional and organic cultivation in olive and sesame oil, coconut, canola, and mustard seed [39,40,41].

The thrombogenicity index (TI) was higher in conventionally grown lemons, and there were no significant differences in the atherogenicity index (ATI). Lower values of TI and ATI indices indicate the fatty acids ratio is more favorable to human health [42]. In this sense, in both samples, these indices presented low values (~0.33 and ~0.19 of ATI and TI, respectively, for conventional lemons and ~0.34 and ~0.15 of ATI and TI, respectively, for organic lemons). The results showed that the fatty acid profile is influenced by agronomics practices during its cultivation.

### 3.2. Antioxidant Activity and Total Polyphenol Content

The antioxidant activity carried out by the ABTS+, DPPH•, and FRAP methods showed no significant differences between organic and conventional farming (Table 3). These results (~1.57, ~3.08, and ~4.16 mmol Trolox L^−1^ for ABTS+, DPPH•, and FRAP, respectively) were similar to those found by González-Molina et al. [43] for the ‘Verna’ variety and by Aguilar-Hernández, Núñez-Gómez, Forner-Giner, Hernández, Pastor-Pérez and Legua [16], in the ‘Fino 49’, ‘Eureka’, ‘Verna’ and ‘Bétera’ varieties. In other citrus fruits and juices, an increase in total antioxidant capacity was observed in conventional growing [44]. On the other hand, in red oranges and other fruits such as apples, blueberries, grapes, kiwis, peaches, or strawberries, the total antioxidant capacity increased under organic farming conditions [45,46]. Polyphenols are secondary metabolites produced as a defensive method against insects, fungi, or mechanical damage. Due to less use of fertilizers, organic crops are expected to have a higher accumulation of these compounds [41]. Nevertheless, there were no differences in terms of TPC (conventional: 85.51 mg GAE 100 mL^−1^; organic: 86.69 mg GAE 100 mL^−1^). These results were similar to those obtained by Aguilar-Hernández et al. [47] in different lemon cultivars, and those reported by Dangour et al. [48] in their study comparing conventional and organic farming in crops.

### 3.3. Amino Acids

Eighteen amino acids were identified in the lemon samples: alanine, arginine, asparagine, aspartic acid, cysteine, glutamate, glutamine, glycine, histidine, isoleucine, leucine, lysine, methionine, phenylalanine, proline, serine, threonine, tryptophan, tyrosine, and valine (Table 4).

Among these amino acids, the main 6 were: alanine (145.97 and 144.36 mg L^−1^, conventional and organic sample, respectively), phenylalanine (81.87 and 80.09 mg L^−1^), aspartic acid (78.62 and 71.18 mg L^−1^), iso-leucine (69.05 and 53.97 mg L^−1^), leucine (56.16 and 39.94 mg L^−1^), and glutamate (58.93 and 55.83 mg L^−1^). The most notable differences were found in isoleucine and leucine, in which the lemons obtained through conventional farming had higher concentrations, 22% and 29%, respectively, than those obtained through organic farming. Similar results were found by Lorente et al. [49] in lemon juices as the main amino acids present in Spanish lemons.

In general, conventional foods have a higher content of free amino acids due to the lower use of nitrogenous fertilizers [50]. However, recent studies seem to affirm that although organic crops contain less protein and amino acids, they are of higher quality than conventional crops [45].

### 3.4. Polyphenols

The lemon fruit is rich in phenolic compounds, especially flavonoids [51], and therefore, they have an important role in the prevention of diseases such as some types of cancer, diabetes, obesity, and cardiovascular disorders [20]. Therefore, it is not surprising that the results showed a high concentration of flavonoids. Among the identified polyphenols, we found 1 hydroxycinnamic acid (derived from coumarin) 2 flavones (vicenin-2 and diosmetin-diglucoside), 3 flavonols (quercetin glucosyl-rhamnosyl-glucoside, myricetin-rutinoside, and rutin), and 3 flavanones (eriocitrin, hesperidin, and neo-hesperidin) (Table 5). Among all the identified compounds, eriocitrin and neo-hesperidin are the two major ones 122.16 mg L^−1^ in the conventional sample and 123.97 mg L^−1^ in the organic sample and 106.34 mg L^−1^ in the conventional sample and 109.35 mg L^−1^ in the organic sample, respectively. However, only the neo-hesperidin showed significant differences. The lemons grown organically had a higher concentration. In case of citric acid, no significant differences were found (Table 5).

In contrast to the results found, Uckoo et al. [52] observed that hesperidin increased in organic lemons. On the other hand, several authors reported an increase of polyphenols in blueberry fruits such as cyanidin 3-galactoside, delphinidin 3-arabinoside, delphinidin 3-galactoside, delphinidin 3-glucoside, quercetin 3-glucoside, myricetin 3-arabinoside, malvidin 3-glucoside, malvidin 3-arabinoside, petunidin 3-galactoside, and petunidin 3-glucoside among others [53,54,55].

### 3.5. Sensory Analysis

The descriptive sensory analysis was made with 16 descriptors and was found to have significant differences in 8 of them (overall aroma, lemon odor, fresh lemon flavor, citric flavor, floral flavor, sweet, sour, and astringent) (Table 6). With respect to the odor attribute, conventional lemon had less overall aroma intensity and lemon odor (4.6 and 5.1, respectively) than organic lemon (6.5 and 6.8, respectively). Among flavor attributes, citric, sweet, sour, floral, fresh lemon, and astringent were significantly affected by growing conditions. Conventional lemon had the highest values for citric, sweet, and astringent attributes (1.6, 2.5, and 4.4, respectively) and the lowest values for sour, floral, and fresh lemon attributes (4.6, 0.1, and 9.0, respectively).

According to Barnes et al. [56], in lemon yogurts, the key attribute in consumer taste is lemon flavor (fresh lemon) and our results showed that conventional and organic farming led to obtaining lemon with a high intensity of this attribute. Low sweetness and high sourness were correlated with overall liking too [56] and were considered as key attributes of lemon quality, highlighting in this case the lemons grown under organic conditions.

No significant statistical differences were found with respect to the bitter attribute on samples under study. Guneser and Yilmaz [57] showed that bitterness was the principal descriptive sensory attribute of lemon seed oil. The bitter and astringent flavors are caused by the flavonoids present in lemon [58].

## 4. Conclusions

The results obtained in this study demonstrated that conventional farming conditions had a higher content of amino acids and fatty acids than lemons obtained by organic farming conditions. Organic lemons had a better sensory profile and less thrombogenicity index. Farming conditions (conventional or organic) had no significant impact on the polyphenol content and antioxidant activity of lemon (‘Fino 49’). However, these small differences do not allow us to conclude that the use of the type of growing conditions (conventional or organic) is a sufficiently determining factor to significantly influence the general composition of the lemon. Therefore, the choice of one system or another must depend on the conditions of each growing area, trying to opt, whenever possible, for systems that favor sustainability and respect for the environment.

## Figures and Tables

**Table 1 foods-12-04304-t001:** Lexicon was used for the descriptive analysis of ‘Fino 49’ lemon juice.

Attributes	Definition	References
Odor		
Overall aroma	Total overall orthonasal aroma	Lemon juice Solimon = 9
Lemon odor	Odor associated with fresh lemon	Lemon juice Solimon = 7; Freshly cut lemon = 10
Flavor		
Fresh lemon	Aroma related to fresh-cut lemon	Freshly cut lemon = 10
Citric	Aroma associated with citral	1 ppm citral = 9; 0.5 ppm citral = 4.5
Floral	Aroma associated with linalool	1 ppm Linalool = 10
Fruity	Aroma associated with fruit	Fruity Pebbles cereals Post = 10
Cleaner	Aroma associated with citrus cleaners	Lemon Pledge = 10
Sweet	Sweet perceptions associated with sweeteners or sucrose	5 g sucrose/100 mL H_2_O = 5; 10 g sucrose/100 mL H_2_O = 10
Sour	Taste associated with acids	0.05 g citric/100 mL H_2_O = 2; 0.08 g citric/100 mL H_2_O = 5
Bitter	The taste associated with caffeine or quinine	0.05 g caffeine/100 mL H_2_O = 2; 0.08 g caffeine/100 mL H_2_O = 5
Other	Aromatic associated with non-citrus fruits	Multifruit juice Juver = 10; Orange juice Juver = 9
Aftertaste	Time elapse until the taste disappears from the mouth	5 s = 1; 10 s = 5; 20 s = 10
Numbing	Numb feeling (lost feeling) in the oral cavity	1 cough drop Hall’s sugar-free honey-lemon/300 mL boiling water = 3
Mouth coating	Sensation of coating that remains in the oral cavity (tongue and teeth included) after expectoration	0.55% *w/v* carboxyl methyl cellulose in deionized H_2_O = 10
Tooth etching	Sensation of dryness (drawing and/or puckering) on the teeth (in contact with the tongue) after expectoration	Welch’s grape juice = 10
Astringent	Sensation of drying in the oral cavity	0.02 g alum/100 mL H_2_O = 2

**Table 2 foods-12-04304-t002:** Fatty acids (mg 100 g*^−^*^1^) of lemon skin ‘Fino 49’ cultivar, farming under conventional and organic practices.

Fatty Acid	ANOVA ^†^	Conventional	Organic
C8:0	NS	0.44	0.47
C10:0	NS	1.30	0.87
C12:0	NS	3.39	2.52
C13:0	NS	1.19	0.65
C14:0	NS	5.62	5.58
C15:0	**	8.44 a ^‡^	2.41 b
C16:0	***	73.36 a	30.28 b
C16:1	NS	2.37	1.68
C17:0	NS	3.80	3.57
C17:1	NS	0.57	0.10
C18:0	**	9.91 a	3.72 b
C18:1t9	NS	2.01	1.19
C18:1c9	**	27.56 a	10.11 b
C18:1c11	NS	5.10	2.31
C18:2n6c	***	126.29 a	60.07 b
C18:3n6 gamma	NS	2.08	1.94
C18:3n3 alfa	***	125.63 a	76.15 b
Σ SFA	***	107.44 a	50.09 b
Σ MUFA	***	37.61 a	15.38 b
Σ PUFA	***	254.00 a	138.16 b
PUFA/SFA	NS	2.36	2.76
ATI	NS	0.33	0.34
TI	**	0.19 a	0.15 b

^‡^ Values followed by the same letter were not significantly different (*p* < 0.05). ^†^ NS: not significant at *p* > 0.05; ** and ***: significant at *p* < 0.01 and 0.001, respectively. Values are the mean of 3 replicates SFA = saturated fatty acids; MUFA = monounsaturated fatty acids; PUFA = polyunsaturated fatty acids. ATI = atherogenicity index; ATI = (C12:0 + 4 × C14:0 + C16:0)/[ΣMUFA + ΣPUFA (*n* − 6) and (*n* − 3)]; TI = thrombogenicity index; TI = (C14:0 + C16:0 +C18:0)/[0.5 × ΣMUFA + 0.5 × ΣPUFA (*n* − 6) + 3 × ΣPUFA (*n* − 3) + (*n* − 3)/(*n* − 6)].

**Table 3 foods-12-04304-t003:** Antioxidant activity (mmol Trolox L*^−^*^1^) and total polyphenol content (mg GAE 100 mL*^−^*^1^) of lemon fruits ‘Fino 49’ cultivar, under conventional and organic farming.

	ABTS+	DPPH•	FRAP	TPC
ANOVA ^†^	NS	NS	NS	NS
Conventional	1.50	3.10	4.30	85.51
Organic	1.64	3.06	4.02	86.69

^†^ NS: not significant at *p* > 0.05. Values are the mean of 3 replicates (*n* = 3). TPC = Total Polyphenol Content.

**Table 4 foods-12-04304-t004:** Amino acids (mg L^−^^1^) of lemon fruits ‘Fino 49’ under conventional and organic farming.

Amino Acids	ANOVA ^†^	Conventional	Organic
Alanine	NS	145.97	144.36
Arginin	***	6.85 a ^‡^	6.00 b
Aspartic acid	***	78.62 a	71.18 b
Cysteine	**	0.21 a	0.20 b
Glutamate	*	58.93 a	55.83 a
Glycine	**	12.77 a	10.81 b
Histidine	***	0.85 a	0.73 b
Isoleucine	***	69.05 a	53.97 b
Leucine	***	56.16 a	39.94 b
Lysine	***	0.85 a	0.72 b
Methionine	*	3.37 a	3.64 a
Phenylalanine	**	81.87 a	80.09 b
Proline	***	19.73 a	12.32 b
Serine	***	47.34 a	41.57 b
Threonine	***	2.79 a	2.65 b
Tryptophan	***	6.76 b	10.24 a
Tyrosine	***	31.35 a	23.74 b
Valine	***	17.47 a	16.50 b
Total	**	640,930 a	574,502 b

^‡^ Values followed by the same letter were not significantly different (*p* < 0.05). ^†^ NS: not significant at *p* > 0.05, *, ** and *** significant at *p* < 0.05, 0.01 and 0.001, respectively. Values are the mean of 3 replicates (*n* = 3).

**Table 5 foods-12-04304-t005:** Polyphenols (ppm) of lemon fruits ‘Fino 49’ under conventional and organic farming.

Polyphenols	m/z-	RT	ANOVA ^†^	Conventional	Organic
Citric Acid	191	3.8	NS	13.04	14.62
Derived from Coumarin	393-325	14.6	***	10.48 a ^‡^	9.21 b
Quercetin glucosyl-rhamnosyl-glucoside	771-609-301	15.8	NS	30.22	30.19
Vicenin-2	593-353-325-298	17.1	**	13.40 b	15.03 a
Myricetin-rutinoside	317-289	17.4	**	14.46 b	15.81 a
Diosmetin-diGlucoside	623-383-312	18.3	NS	79.50	81.24
Eriocitrin	595-287-269	19.7	NS	122.16	123.97
Rutin	609-301	20.9	NS	11.61	12.79
Hesperidin	609-301	22.5	NS	9.26	9.63
Neo-hesperidin	609-301	23.2	**	106.34 b	109.35 a
Total			NS	410.45	421.82

^‡^ Values followed by the same letter were not significantly different (*p* < 0.05). ^†^ NS: not significant at *p* > 0.05, ** and *** significant at *p* < 0.01 and 0.001, respectively. Values are the mean of 3 replicates (*n* = 3). RT = Retention time.

**Table 6 foods-12-04304-t006:** Sensory descriptive test on conventional and organic lemon juice.

Attributes	ANOVA ^†^	Conventional	Organic
Odor			
Overall aroma	**	4.6 b ^‡^	6.5 a
Lemon odor	***	5.1 b	6.8 a
Flavor			
Citral	*	1.6 a	1.3 b
Sweet	***	2.5 a	1.6 b
Fruity	NS	0.2	0.3
Sour	***	4.6 b	5.8 a
Bitter	NS	5.1	4.9
Cleaner	NS	1.6	1.5
Floral	**	0.1 b	0.6 a
Fresh lemon	***	9.0 b	9.8 a
Other	NS	0.1	0.1
Aftertaste	NS	6.8	6.6
Numbing	NS	4.6	4.8
Mouth coating	NS	2.2	2.2
Tooth etching	NS	1.5	1.3
Astringent	***	4.4 a	4.0 b

^‡^ Values followed by the same letter were not significantly different (*p* < 0.05). ^†^ NS: not significant at *p* > 0.05; *, ** and ***: significant at *p* < 0.05, 0.01 and 0.001, respectively.

## Data Availability

Data is contained within the article.

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
