# Peer review of "Conventional vs. Organic: Evaluation of Nutritional, Functional and Sensory Quality of *Citrus limon"

_foods, 2023, doi:10.3390/foods12234304_

Round 1
Reviewer 1 Report
Comments and Suggestions for Authors
The paper “Conventional vs organic: evaluation of nutritional, functional and sensory quality of Citrus limon‘’ aimed to evaluate the nutritional, sensorial and functional quality of the ‘Fino 49’ lemon grafted on Citrus macrophylla in conventional and organic cultivation
As authors indicated clearly nowadays, consumers value fruit not only for their visual appeal, but also for their nutritional and functional attributes. In this sense, organic agriculture is highly appreciated by consumers, since consumers around the world believe that organic food has a higher content of beneficial compounds for health and consider it of higher quality.
The paper is prepared professionally. It includes a well-crafted abstract and an exhaustive introduction that justifies the research undertaken. The introduction points to the deficiencies in the literature on the subject. The aim is clearly defined. Modern analytical methods were used in the research. The discussion of the results is well prepared. The conclusions are well-defined. The illustrative material is appropriate.
In my opinion, the manuscript after corrections, will be better.
Detailed comments:
Abstract:
Line 12-17 general description is so long and must be shortened or move some part to introduction.
Abstract must include some more numeric data obtained from the study
Do not use abbreviations when use first time.
Abstract should be rewritten based on international rules.
Line 22. of ascorbic acid (1.96 mg L-1)..whey Vitamin C is so lower as 0.19 mg per 100 g????? any analytical mistake?????
Line 24...organic citrus is not correct use organically produced lemon fruits
Please add some sentence at the and of abstract about future scenario based on the results!!!!!!
Introduction - The introduction is enough in my opinion. Introduction needs some minor changes
Line 37 chemical agriculture is not true use conventional agriculture
Line 49-50 In general, organic agriculture presents soils with a higher content of organic matter and greater biodiversity [10]....this is repeat of previous sentences!!!!
Line 71-73 Currently, the quality of the fruit is valued, not only at the visual level (size, color), but also at the nutritional and functional level (content of carotenoids, minerals, phenols, vitamins and volatile compounds, etc.) [14,20]. Please add more references I can suggest below ones.
(Ozrenk et al., 2020; Abanoz and Okcu, 2022; Dawadi et al., 2022).
Ozrenk, K., Ilhan, G., Sagbas, H. I., Karatas, N., Ercisli, S., Colak, A. M.(2020). Characterization of European cranberrybush (Viburnum opulus L.)genetic resources in Turkey. Scientia Horticulturae, 273, 109611
Dawadi, P., Shrestha, R., Mishra, S., Bista, S., Raut, R.K., Joshi, T.P., Bhatt, L.R., 2022. Nutritional value and antioxidant properties of Viburnum mullaha Buch.-Ham. Ex D. Don fruit from central Nepal. Turkish Journal of Agriculture and Forestry, 46 (5), 781-789.
Abanoz YY, Okcu Z (2022). Biochemical content of cherry laurel (Prunus laurocerasus L.) fruits with edible coatings based on caseinat, Semperfresh and lecithin. Turkish Journal of Agriculture and Forestry, 46 (6), 908-918.
Rest of parts are OK
Comments on the Quality of English LanguageNo comment
Author Response
The paper “Conventional vs organic: evaluation of nutritional, functional and sensory quality of Citrus limon‘’ aimed to evaluate the nutritional, sensorial and functional quality of the ‘Fino 49’ lemon grafted on Citrus macrophylla in conventional and organic cultivation
As authors indicated clearly nowadays, consumers value fruit not only for their visual appeal, but also for their nutritional and functional attributes. In this sense, organic agriculture is highly appreciated by consumers, since consumers around the world believe that organic food has a higher content of beneficial compounds for health and consider it of higher quality.
The paper is prepared professionally. It includes a well-crafted abstract and an exhaustive introduction that justifies the research undertaken. The introduction points to the deficiencies in the literature on the subject. The aim is clearly defined. Modern analytical methods were used in the research. The discussion of the results is well prepared. The conclusions are well-defined. The illustrative material is appropriate.
In my opinion, the manuscript after corrections, will be better.
The authors thank the reviewers for their helpful comments. All the suggestions have been considered and the manuscript has been improved by incorporating reviewers’ comments/suggestions.
Line 12-17 general description is so long and must be shortened or move some part to introduction.
The introduction part of the abstract has been shortened. Please, see lines 12-15.
Abstract must include some more numeric data obtained from the study
Included as suggested. Please, see lines 22-26.
Do not use abbreviations when use first time.
Manuscript has been reviewed and abbreviations are described.
Abstract should be rewritten based on international rules.
Abstract has been rewritten.
Line 22. of ascorbic acid (1.96 mg L-1)..whey Vitamin C is so lower as 0.19 mg per 100 g????? any analytical mistake?????
Reviewer is absolutely right. Results of Vit-C are wrong. As the authors can not be sure of the results, we are decided to remove it.
Line 24...organic citrus is not correct use organically produced lemon fruits
Modified as suggested. Please, see line 21
Please add some sentence at the and of abstract about future scenario based on the results!!!!!!
Added as suggested. Please, see lines 26-27
Introduction - The introduction is enough in my opinion. Introduction needs some minor changes
Line 37 chemical agriculture is not true use conventional agriculture
The reviewer is completely right. The authors apologize for the mistake. The sentence has been modified. Please, see line 38.
Line 49-50 In general, organic agriculture presents soils with a higher content of organic matter and greater biodiversity [10]....this is repeat of previous sentences!!!!
Sentence has been rewritten. Please, see lines 50-52.
Line 71-73 Currently, the quality of the fruit is valued, not only at the visual level (size, color), but also at the nutritional and functional level (content of carotenoids, minerals, phenols, vitamins and volatile compounds, etc.) [14,20]. Please add more references I can suggest below ones.
(Ozrenk et al., 2020; Abanoz and Okcu, 2022; Dawadi et al., 2022).
Ozrenk, K., Ilhan, G., Sagbas, H. I., Karatas, N., Ercisli, S., Colak, A. M.(2020). Characterization of European cranberrybush (Viburnum opulus L.)genetic resources in Turkey. Scientia Horticulturae, 273, 109611
Dawadi, P., Shrestha, R., Mishra, S., Bista, S., Raut, R.K., Joshi, T.P., Bhatt, L.R., 2022. Nutritional value and antioxidant properties of Viburnum mullaha Buch.-Ham. Ex D. Don fruit from central Nepal. Turkish Journal of Agriculture and Forestry, 46 (5), 781-789.
Abanoz YY, Okcu Z (2022). Biochemical content of cherry laurel (Prunus laurocerasus L.) fruits with edible coatings based on caseinat, Semperfresh and lecithin. Turkish Journal of Agriculture and Forestry, 46 (6), 908-918.
References add as suggested. Please, see lines 74, and 384-390
Reviewer 2 Report
Comments and Suggestions for Authors
This study aimed to evaluate the nutritional, sensorial and functional quality of the ‘Fino 49’ lemon grafted on Citrus macrophylla in conventional and organic cultivation. Fatty acids, amino acids, total phenol, and polyphenols were quantified, antioxidant activity was measured, and sensory descriptive analysis was performed. However, there are some weaknesses that need to be addressed.
1. Line 12: suggest to change “due to is perceived” to “since it is perceived”.
2. Introduction:
Please add the content about why do you choose the Fino 49 lemon? Should include more background on the nutritional, sensorial and functional properties of lemon or Fino 49 lemon.
3. Line 131: suggest to change “dilute” to “diluted”, change “dilute” to “centrifuge and filter” to “centrifuged and filtered”.
4. Line 221: how do you get the data of ~1.60, ~3.08, and ~4.16 mmol Trolox L-1 for ABTS+, DPPH• and FRAP. The results were different with the Table 3.
5. Table 3: the data of the critic acid in lemon fruits were not listed in Table 3. Should delete “citric acid (mg L-1)” in the table title.
Expand TPC in the table footnote
6. Line 233-234: add the abbreviation of total polyphenol content (TPC) on the first appearance in the article.
The results of total polyphenol content (conventional: 285.51 mg GAE 100 mL-1; organic: 286.69 mg GAE 100 mL-1) in the article were different with the data in Table 3. Please check the data.
7. Line 234-237: The discussion of results can be improved. The authors should provide reasoning for results obtained. Add the explanation of the obtained data compared with those in the reported literatures.
8. Line 289: suggest to change “Polyfphenols” to “Polyphenols”.
9. Table 5: The results of citric acid were shown in Table 5, why are there two groups of data in the results. Add the discussion of citric acid in the article.
Expand RT in the table footnote.
Comments on the Quality of English Language
minor editing of English language required
Author Response
This study aimed to evaluate the nutritional, sensorial and functional quality of the ‘Fino 49’ lemon grafted on Citrus macrophylla in conventional and organic cultivation. Fatty acids, amino acids, total phenol, and polyphenols were quantified, antioxidant activity was measured, and sensory descriptive analysis was performed. However, there are some weaknesses that need to be addressed.
We thank reviewer comments and suggestions that may allow improving the quality of the paper, we have addressed all comments and below you will find detailed responses to all queries.
- Line 12: suggest to change “due to is perceived” to “since it is perceived”.
Changed as suggested. Please see line 12
- Introduction:
Please add the content about why do you choose the Fino 49 lemon? Should include more background on the nutritional, sensorial and functional properties of lemon or Fino 49 lemon.
Included as suggested. Please, see lines 63-66.
- Line 131: suggest to change “dilute” to “diluted”, change “dilute” to “centrifuge and filter” to “centrifuged and filtered”.
Changed as suggested. Please, see line 135.
- Line 221: how do you get the data of ~1.60, ~3.08, and ~4.16 mmol Trolox L-1for ABTS+, DPPH• and FRAP. The results were different with the Table 3.
The results described in the text are the average values obtained in each test, taking into account the factor studied. We decided to express it this way since there are no significant differences, and thus the text is easier to read.
- Table 3: the data of the critic acid in lemon fruits were not listed in Table 3. Should delete “citric acid (mg L-1)” in the table title.
The reviewer is completely correct. The authors apologize for the error. The table title is now correct. Please, see lines 241-242.
Expand TPC in the table footnote
Done as suggested. Please, see line 245.
- Line 233-234: add the abbreviation of total polyphenol content (TPC) on the first appearance in the article.
Changed for the abbreviation. Please, see line 236.
The results of total polyphenol content (conventional: 285.51 mg GAE 100 mL-1; organic: 286.69 mg GAE 100 mL-1) in the article were different with the data in Table 3. Please check the data.
The reviewer was right. It was a mistake, the real data are those that appear in the table. Now it is correct in the results and discussion section. Please, see lines 237-238.
- Line 234-237: The discussion of results can be improved. The authors should provide reasoning for results obtained. Add the explanation of the obtained data compared with those in the reported literatures.
Added as suggested. Please, see lines 238-240.
- Line 289: suggest to change “Polyfphenols” to “Polyphenols”.
It was a mistake, the word has been corrected. Please, see line 292.
- Table 5: The results of citric acid were shown in Table 5, why are there two groups of data in the results. Add the discussion of citric acid in the article.
Citric acid was found as 2 isomers at different retention times. For this reason, initially, there were 2 concentration values in the table. However, after the reviewer's valuable comment, it was decided to group the results. Please see Table 5. A short line has also been added indicating that no statistically significant differences were found for this polyphenol. Please, see line 284.
Expand RT in the table footnote.
Done as suggested. Please, see line 295.
Reviewer 3 Report
Comments and Suggestions for Authors
I suggest reconsideration of the paper after major revision. The comments are following:
Abstract:
The individual results in abstract are well presented, but the general conclusion is omitted. It is not clear what is the main conclusion obtained in this research. This is rather to be in abstract part than the particular results that are already presented in the part Results and discussion. This could be overlooked in case that the general conclusion is presented in the Conclusion part, but also it is not the case with this paper.
Introduction:
The main complaints in the introduction part are the following:
- The text is too long and confusing text that contains a lot of information that is not clearly organized. Because of this, for the first reading it is difficult to follow the main objectives of the research.
- Although the aim of the research is stated in the last paragraph of the paper, it is not clearly stated what is the hypotheses that is investigated and also it is not clearly stated why this research is significant and what problems it is trying to resolve.
Materials and Methods:
2.1. Plant Material – It is not described in which way the material is prepared for analysis.
2.2. Fatty Acids – The font of this chapter is different from the rest of the text.
The description of the Thrombogenicity index (TI) analyse was not described in the part Materials and Methods.
The organization of the part Materials and Methods requires complete reorganization, since it is unreviewed and disorganized. If each analyse is described in each paragraph separately, the text would be more clear.
Results and Discussion:
The same as in the part Materials and Methods, this part is not well organized and it requires seriously reorganization with the clear subtitles of the chapters and to separate the results to be clearer and understand for reading.
Conclusion:
Conclusion part requires addition of the general conclusion. Currently the conclusion is just three sentences taken from the part of the part Results and Discussion and they do not indicate on specific value of the research and they not explain why the research was conducted.
Comments on the Quality of English Language
Moderate editing of English language required.
Author Response
The individual results in abstract are well presented, but the general conclusion is omitted. It is not clear what is the main conclusion obtained in this research. This is rather to be in abstract part than the particular results that are already presented in the part Results and discussion. This could be overlooked in case that the general conclusion is presented in the Conclusion part, but also it is not the case with this paper.
- The text is too long and confusing text that contains a lot of information that is not clearly organized. Because of this, for the first reading it is difficult to follow the main objectives of the research.
- Although the aim of the research is stated in the last paragraph of the paper, it is not clearly stated what is the hypotheses that is investigated and also it is not clearly stated why this research is significant and what problems it is trying to resolve.
Thank you very much for your comment. Changes have been made to the introduction and conclusion to facilitate reading comprehension.
Materials and Methods:
2.1. Plant Material – It is not described in which way the material is prepared for analysis.
Added as suggested. Please, see lines 97-99.
2.2. Fatty Acids – The font of this chapter is different from the rest of the text.
The reviewer is totally right. The font has been modified. Please, see lines 101-115.
The description of the Thrombogenicity index (TI) analyse was not described in the part Materials and Methods.
The thrombogenicity index is calculated as: TI= (C14:0 + C16:0 +C18:0)/ [0.5 x ΣMUFA + 0.5 x ΣPUFA (n-6) + 3 x ΣPUFA (n-3) + (n- 3)/(n-6)]. This information is included in the footer of Table 2.
The organization of the part Materials and Methods requires complete reorganization, since it is unreviewed and disorganized. If each analyse is described in each paragraph separately, the text would be more clear.
Results: The same as in the part Materials and Methods, this part is not well organized and it requires seriously reorganization with the clear subtitles of the chapters and to separate the results to be clearer and understand for reading.
Done as suggested. The analyses have been separated. Please, see Materials and Methods and Results sections.
Conclusion:
Conclusion part requires addition of the general conclusion. Currently the conclusion is just three sentences taken from the part of the part Results and Discussion and they do not indicate on specific value of the research and they not explain why the research was conducted.
Conclusion has been rewritten.
Reviewer 4 Report
Comments and Suggestions for Authors
the paper entitled 'Conventional vs organic: evaluation of nutritional, functional 2 and sensory quality of Citrus limon' is dealing with an important topic which is conventional and organic farming effect on Citrus lemon.
The paper generally is fine from the abstract to the conclusion but only two small comments
1/The paper lacks also preferably an ACP resuming all the traits analysed within the paper
2/The conclusion needs also to be further improved and extended
Comments on the Quality of English LanguageOnly minor editing is required
Author Response
The paper entitled 'Conventional vs organic: evaluation of nutritional, functional 2 and sensory quality of Citrus limon' is dealing with an important topic which is conventional and organic farming effect on Citrus lemon.
We thank reviewer comments and suggestions that may allow improving the quality of the paper, we have addressed all comments and below you will find detailed responses to all queries.
The paper generally is fine from the abstract to the conclusion but only two small comments
1/The paper lacks also preferably an ACP resuming all the traits analysed within the paper
The authors appreciate the reviewer's comment. Yes, a principal component analysis could be helpful to the reader. However, this type of analysis cannot be carried out with only 2 variables. The software allows working from 3 variables, and the protocols recommend that for the results to be truly useful, work with at least 5.
2/The conclusion needs also to be further improved and extended
Conclusion has been rewritten.
Round 2
Reviewer 1 Report
Comments and Suggestions for Authors
This research sought to assess the nutritional, sensory, and functional attributes of 'Fino 49' lemons grafted on Citrus macrophylla in both conventional and organic cultivation settings. The study involved the quantification of fatty acids, amino acids, total phenols, and polyphenols, alongside the measurement of antioxidant activity, and a sensory descriptive analysis. Nevertheless, there exist certain shortcomings that warrant attention and rectification.
Reviewer 2 Report
Comments and Suggestions for Authors
accept in the present form.